# Characterizing Missing Information in Deep Networks Using Backpropagated Gradients

## Abstract

Deep networks face challenges of ensuring their robustness against inputs that cannot be effectively represented by information learned from training data. We attribute this vulnerability to the limitations inherent to activation-based representation. To complement the learned information from activation-based representation, we propose utilizing a gradient-based representation that explicitly focuses on *missing information*. In addition, we propose a directional constraint on the gradients as an objective during training to improve the characterization of missing information. To validate the effectiveness of the proposed approach, we compare the anomaly detection performance of gradient-based and activation-based representations. We show that the gradient-based representation outperforms the activation-based representation by $0.093$ in CIFAR-10 and $0.361$ in CURE-TSR datasets in terms of AUROC averaged over all classes. Also, we propose an anomaly detection algorithm that uses the gradient-based representation, denoted as GradCon, and validate its performance on three benchmarking datasets. The proposed method outperforms the majority of the state-of-the-art algorithms in CIFAR-10, MNIST, and fMNIST datasets with an average AUROC of $0.664$, $0.973$, and $0.934$, respectively.

## 1 Introduction

The generalizable representation of data from deep network has largely contributed to the success of deep learning in diverse applications (Bengio et al., 2013). The representation from deep networks is often obtained in the form of activation. The activation is constructed by the weights which contain specific knowledge learned from training samples. Recent studies reveal that deep networks still face robustness issues when input that cannot be properly represented by learned knowledge is given to the networks (Goodfellow et al., 2014; Hendrycks & Dietterich, 2018; Liang et al., 2017). One of the reasons for the vulnerability of deep networks is the limitation in the activation-based representation, which inherently focused on the learned knowledge. However, the part of the input that causes problems in deep networks is mainly from the information that deep networks were not able to learn from the training data. Therefore, it is more appropriate to complement the representation of input data from the perspective of information that has not been learned for enhancing the robustness of machine learning algorithms.

The gradient is another fundamental element in deep networks that is utilized to learn new information from given inputs by updating model weights (Goodfellow et al., 2016). It is generated through backpropagation to train deep networks by minimizing designed loss functions (Rumelhart et al., 1986). During the training of network, the gradient with respect to the weights provides directional information to update the deep network and learn a better representation for the inputs. In other words, gradients guide the network to learn new information that was not learned from data that it has seen so far but is presented in the current input. Considering this role during training, gradients can provide a complementary perspective with respect to activation and characterize missing information that the network has not learned for each unseen image.

We demonstrate the role of gradients with an example in Fig. 1. Assume that a deep network has only learned curved edge features from training images of the digit '0'. During testing, the digit '6' is given to the network. The digit '6' consists of both learned information (curved edges) and missing information (straight edges on top). Since the activation-based representation is constructed

based on the information that the network has already learned, the curved part of the digit '6' will be characterized effectively by the activation. However, the network still has to learn the straight edge features to perform successfully on the digit '6'. Therefore, the gradients which guide updates in the deep network can characterize straight edge information that has not been learned.

We propose analyzing the representation capability of gradients in characterizing missing information for deep networks. Gradients have been utilized in diverse applications such as adversarial attack generation and visualization (Zeiler & Fergus, 2014; Goodfellow et al., 2014). However, using gradients with respect to weight as the representation of data has not been actively explored yet. Through the comprehensive analysis with activation-based representations, we show the effectiveness of gradient representation in characterizing the information that has not been learned for deep network. Furthermore, we show that gradient representation can achieve state-of-the-art performance in detecting potentially invalid data for the network. The main contributions of this paper are three folds:

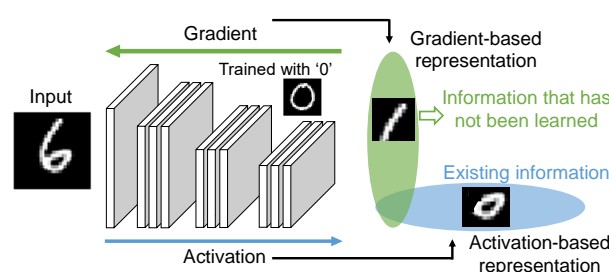

Figure 1: Gradient-based representation for characterizing information that has not been learned for the deep network.

i We propose utilizing gradients as a representation to characterize information that has not been learned from the training data but is currently presented in the input data.

ii We analyze the representation capability of gradient compared to activation for detecting samples which possess features that have not been learned for the network.

iii We propose a gradient-based anomaly detection algorithm that outperforms state-of-the-art algorithms based on activation representations.

## 2 RELATED WORKS

Existing works have focused on achieving reliable activation-based representations to properly handle inputs that can cause significant performance degradation to deep networks. One of the most intuitive ways is to enhance the representation capability of activation by finetuning trained models and learning more from augmented data. Goodfellow et al. (2014); Vasiljevic et al. (2016); Temel et al. (2017) utilize adversarial images, blurred images, and distorted virtual images, respectively, to finetune networks to improve the classification performance based on the activation features. Also, several pre-processing techniques have been developed to make the representation of data similar to that of training data to achieve the robustness of algorithms. Discrete cosine transform (DCT) has been explored as a simple pre-processing technique to eliminate the effect of distortion and adversarial attacks (Hossain et al., 2018; Das et al., 2018). On the other hand, methods developed for detecting and filtering out problematic samples have focused on making the representation of such samples as distinguishable as possible from that of training images in the activation space. Liang et al. (2017) propose a gradient-based pre-processing technique to make the activation-based representation of adversarial images and out-of-distribution images dissimilar to that of training images. Furthermore, constrained activation representations have been actively utilized to make normal and abnormal images statistically separable in the activation space for abnormal data detection (Pidhorskyi et al., 2018; Perera et al., 2019; Abati et al., 2018). Aforementioned works exclusively focus on activation-based representations which are based on learned information to handle the information that has not been learned. We complement these by proposing backpropagated gradients-based representation of data which particularly focuses on what has not been learned in the trained network.

The backpropagated gradients have been utilized in diverse applications including but not limited to visualization, adversarial attacks, and image classification. The backpropagated gradients have been widely used for the visualization of deep networks. In Zeiler & Fergus (2014); Springenberg et al. (2014), information that networks have learned for a specific target class is mapped back to the pixel space through the backpropagation and visualized. Selvaraju et al. (2017) utilize the

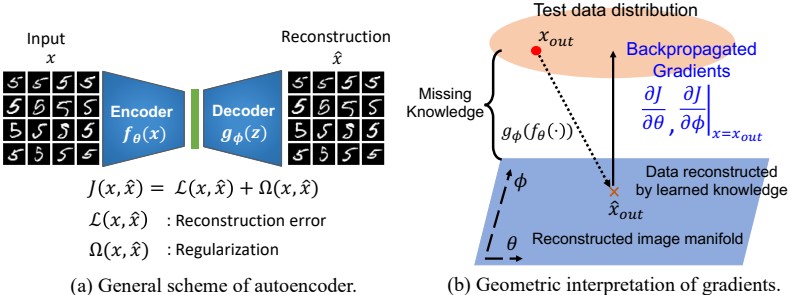

(a) General scheme of autoencoder.     (b) Geometric interpretation of gradients.

Figure 2: Geometric visualization for weight gradients from test images.

gradient with respect to activation to weight the activation and visualize the reasoning for prediction that deep networks have made. Adversarial attack is another application of gradients. Goodfellow et al. (2014); Kurakin et al. (2016) show that adversarial attacks can be generated by adding an imperceptibly small vector which is the signum of input gradients. In Kwon et al. (2019), the backpropagated gradients are utilized to classify and objectively estimate the quality of distorted images. Several works have incorporated gradients with respect to input in the form of regularization during the training of deep networks to improve the robustness (Drucker & Le Cun, 1991; Ross & Doshi-Velez, 2018; Sokolić et al., 2017). Although existing works have shown that gradients with respect to input or activation can be useful for diverse applications, gradients with respect to weight remain almost unexplored aside from its role in training deep networks. In the following section, we demonstrate the interpretation of weight gradients to further elaborate the role of weight gradients as data representation.

## 3   GEOMETRIC INTERPRETATION OF GRADIENTS

We use an autoencoder which is an unsupervised representation learning framework to explain the geometric interpretation of gradients. An autoencoder consists of an encoder, $f_\theta$, and a decoder, $g_\phi$ as shown in Fig. 2 (a). From an input image, $x$, the latent variable, $z$, is generated as $z = f_\theta(x)$ and the reconstructed image is obtained by feeding the latent variable into the decoder, $g_\phi(f_\theta(x))$. The training is performed by minimizing the loss function, $J(x; \theta, \phi)$, defined as follows:

$$J(x; \theta, \phi) = \mathcal{L}(x, g_\phi(f_\theta(x))) + \Omega(z; \theta, \phi), \tag{1}$$

where $\mathcal{L}$ is a reconstruction error which measures the dissimilarity between the input and the reconstructed image and $\Omega$ is a regularization term for the latent variable.

We visualize the geometric interpretation of backpropagated gradients in Fig. 2 (b). The autoencoder is trained to accurately reconstruct training images and the reconstructed training images form a manifold. We assume that the structure of the manifold is a linear plane which is spanned by the weights of the encoder and the decoder as shown in the figure for the simplicity of explanation. Also, we explains the generalization of this concept in the following section. During test time, any given input to the autoencoder is projected onto the reconstructed image manifold through the projection, $g_\phi(f_\theta(\cdot))$. The closer test images are to the reconstructed image manifold, the more accurately reconstructed they are. Let us assume that test data distribution is outside of the reconstructed image manifold. When the test image, $x_{out}$, sampled from the test data distribution is given to the autoencoder, it will be reconstructed as $\hat{x}_{out}$ by the projection, $g_\phi(f_\theta(x_{out}))$. Since test images from the test data distribution have not been utilized for training, they will be poorly reconstructed. Given that the reconstructed image manifold contains data reconstructed by learned knowledge, the orthogonal gap between $x_{out}$ and $\hat{x}_{out}$ measures the missing knowledge that the autoencoder has not learned and cannot reconstruct. Using the error defined in (1), the gradient of weights, $\frac{\partial J}{\partial \theta}, \frac{\partial J}{\partial \phi}$, can be calculated through the backpropagation. These gradients represent required changes in the reconstructed image manifold to incorporate the test images and reconstruct them accurately. In other words, these gradients characterize orthogonal variations of the test distribution with respect to the reconstructed image manifold, which is missing information in the trained networks.

The gradients complement activation-based representation in characterizing missing information. In this setup of the autoencoder, missing information is characterized by the reconstruction error using activation. It provides distance information between test images and reconstructed images and is utilized as a primal loss to generate gradients. However, it does not provide any directional informa-

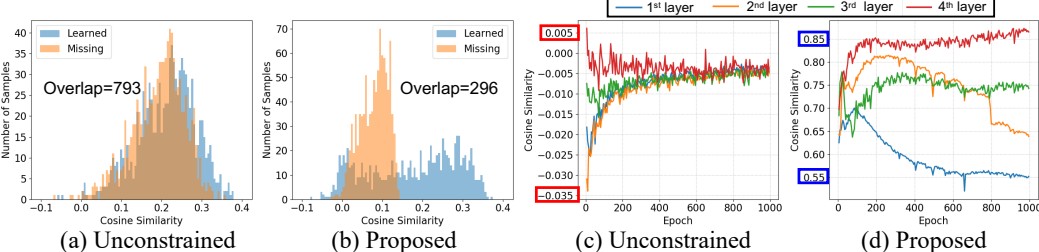

Figure 3: Analysis on the discriminant capability of gradients for learned and missing information.

tion about the deviation of test data distribution. The gradients focus on the directional information to characterize information that has not be learned. Considering that weights contain learned knowledge from training samples, the weight gradients indicate the directional changes required to obtain new knowledge with respect to current knowledge of networks. Therefore, the gradients can provide a comprehensive perspective to represent missing knowledge in trained networks by complementing distance information from activation-based representations with the directional information.

## 4 GENERALIZATION OF THE GEOMETRIC INTERPRETATION FOR GRADIENTS

The gradients require a discriminant capability in high dimensional space to be an effective representation for missing information in practical scenarios of deep networks. As a representation for the information that has not been learned, the representation is expected to clearly differentiate information that has been learned and has not been learned from the input. Therefore, gradients from images with learned information should be distinguishable in terms of direction from those generated by images with new information. For instance, the gradient generated from $x_{out}$ in Fig. 2 is orthogonal to the reconstructed image manifold. However, test images with learned information do not result in high reconstruction errors and do not require significant changes in the reconstructed image manifold. In this case, gradients from these test images will be more tangential to the manifold and distinguishable from the gradient of $x_{out}$. This separation between gradients from learned and not learned information is required to effectively characterize what is missing and not in deep networks.

In addition, gradients from the learned information should be constrained to make the direction of gradients for missing information more distinctive. In Fig. 2, the reconstructed image manifold is a two dimensional plane in three dimensional space. The gradients generated by test images with learned information are tangential to the manifold and constrained to the two dimensional space. These constrained gradients for test images with learned information allow gradients from $x_{out}$ to be more distinctive and characterize missing information in the network.

We design two experiments to analyze the separation between gradients from learned and missing information in the deep network. We train a convolutional autoencoder (CAE) which consists of 4-layer encoder and decoder, respectively. We use training images from `Airplane` class in CIFAR-10 dataset (Krizhevsky & Hinton, 2009) to train the CAE. A test set contains 1000 images from `Airplane` class and the same number of images randomly sampled from all other 9 classes in test split. The test `Airplane` class images contain learned knowledge and images from other classes contain information that has not been learned. In the first experiment, we extract the backpropagated gradients for all test images and measure the average alignment of gradients using cosine similarity (cosSIM). In Fig. 3 (a), we visualize the distribution of the average cosine similarity calculated between the gradient from each `Airplane` class image and that from all other remaining `Airplane` class images (Learned), and between each `Airplane` class image and test images from all other classes (Missing). We calculate the average cosine similarity over all 4 layers in decoder. Ideally for gradients to be an effective representation, cosine similarity between images with learned information should be clearly separable to that calculated between images with learned and missing information. We also calculated the number of samples in the overlapped region of histograms. A large overlapped region (793 samples out of 1000 samples) between cosine similarity from 'Learned' and 'Missing' indicates that gradients in general cannot be distinguishable in high dimensional space. However, with our proposed method which will be described in the following section, the number of overlapped samples significantly decreases from 793 to 296 as shown in

| Model | Loss | Plane | Car | Bird | Cat | Deer | Dog | Frog | Horse | Ship | Truck | Average |
|-------|------|-------|-----|------|-----|------|-----|------|-------|------|-------|---------|
| CAE | Recon | 0.689 | 0.356 | 0.639 | 0.592 | 0.676 | **0.621** | 0.504 | 0.499 | 0.716 | 0.390 | 0.568 |
| CAE + Grad | Recon | 0.659 | 0.356 | **0.640** | 0.555 | 0.695 | 0.554 | 0.549 | 0.478 | 0.695 | 0.357 | 0.554 |
| | Grad | **0.752** | 0.619 | 0.622 | 0.580 | 0.705 | 0.591 | 0.683 | **0.576** | **0.774** | **0.709** | **0.661** |
| VAE | Recon | 0.553 | 0.608 | 0.437 | 0.546 | 0.393 | 0.531 | 0.489 | 0.515 | 0.552 | 0.631 | 0.526 |
| | Latent | 0.634 | 0.442 | **0.640** | 0.497 | **0.743** | 0.515 | **0.745** | 0.527 | 0.674 | 0.416 | 0.583 |
| VAE + Grad | Recon | 0.556 | 0.606 | 0.438 | 0.548 | 0.392 | 0.543 | 0.496 | 0.518 | 0.552 | 0.631 | 0.528 |
| | Latent | 0.586 | 0.396 | 0.618 | 0.476 | 0.719 | 0.474 | 0.698 | 0.537 | 0.586 | 0.413 | 0.550 |
| | Grad | 0.736 | **0.625** | 0.591 | **0.596** | 0.707 | 0.570 | 0.740 | 0.543 | 0.738 | 0.629 | 0.647 |

Table 1: Anomaly detection results based on activations and gradients in CIFAR-10.

Fig. 3 (b). In addition, gradients become more aligned for learned information by achieving high cosine similarity and become orthogonal for missing information. This shows that gradients become more separable for learned and missing information with the proposed approach.

In the second experiment, we analyze the constraint on gradients by measuring the alignment of them generated while the training of CAE. We train the same architecture of the CAE described in the first experiment using `Airplane` class images. To measure the alignment of training gradients, we calculate the cosine similarity between the gradients of a certain layer $i$ at the $k$ th iteration of training, $\frac{\partial \mathcal{L}}{\partial W_i}^k$, and the average of training gradients of layer $i$ obtained until the $k-1$ th iteration, $\frac{\partial \mathcal{L}}{\partial W_i}_{avg}^{k-1}$, defined as follows:

$$\text{cosSIM}\left(\frac{\partial \mathcal{L}}{\partial W_i}_{avg}^{k-1}, \frac{\partial \mathcal{L}}{\partial W_i}^k\right) \quad \text{where} \quad \frac{\partial \mathcal{L}}{\partial W_i}_{avg}^{k-1} = \frac{1}{(k-1)}\sum_{t=1}^{k-1}\frac{\partial \mathcal{L}}{\partial W_i}^t, \quad (2)$$

$W$ is the weight of autoencoder, $i$ is the layer index, and superscripts are used to indicate the iteration number. We visualize the progress of the cosine similarity while training over 1000 epochs in Fig. 3 (c). We particularly focus on the cosine similarity in the 4 layers of decoder. To enhance the distinction between gradients from learned information and missing information, training gradients are expected to be constrained and aligned at the end of training by achieving high cosine similarity. However, we observe that the cosine similarity converges to 0 at the end of training. This indicates that even when most of information in training images is learned, the gradients is orthogonal to the average training gradients and remain unconstrained. On the other hand, we show our proposed method effectively constrains the gradients while training and obtain around $0.55 \sim 0.85$ cosine similarity in Fig. 3 (d). This constraint leads to the clear separability observed in Fig. 3 (b). In following section, we explain the details of our proposed method.

## 5 DIRECTIONAL CONSTRAINT FOR GRADIENTS

We propose a directional constraint that enables gradients to characterize missing information in deep networks. In particular, we use the cosine similarity as a measure of the gradient alignment and constrain the direction of gradients in the form of regularization term in a loss function. At $k$ th iteration of training, the entire loss function is defined as follows:

$$J(x;W) = \mathcal{L}(x;W) - \alpha \mathop{\mathbb{E}}_{i}\left[\text{cosSIM}\left(\frac{\partial \mathcal{L}}{\partial W_i}_{avg}^{k-1}, \frac{\partial \mathcal{L}}{\partial W_i}^k\right)\right], \quad (3)$$

where $\mathcal{L}(x;W)$ is the primal loss for a given task and $\alpha$ is the weight for the gradient constraint, which is the average of cosine similarity calculated for each layer. We apply this constraint on a subset of all layers. Also, we set sufficiently small $\alpha$ value to ensure that gradients actively explore the optimal weights until the primal loss becomes small enough. During the training, $\mathcal{L}(x;W)$ is first calculated from the forward propagation. Through the backpropagation, $\frac{\partial \mathcal{L}}{\partial W_i}^k$ is obtained without updating the weights. Based on the obtained gradient, the entire loss $J$ is calculated and finally the weights are updated using backpropagated gradients from the loss $J$.

We apply this gradient constraint to the autoencoders to evaluate its performance in characterizing information that has not been learned. We only constrain the gradients from 4 layers in decoder and

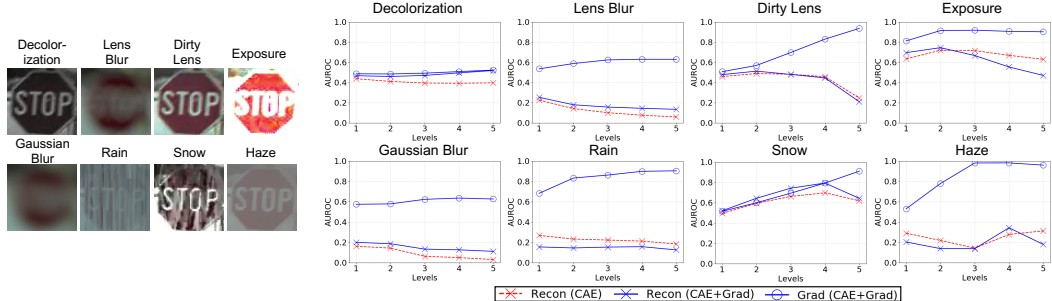

Figure 4: Anomaly detection results based on activations and gradients in CURE-TSR.

encoder layers remain unconstrained. In the setup of autoencoder, the loss can be defined as,

$$J(x; \theta, \phi) = \mathcal{L}(x, g_\phi(f_\theta(x))) + \Omega(z; \theta, \phi) - \alpha \mathop{\mathbb{E}}_{i \in \text{decoder}} \left[ \text{cosSIM} \left( \frac{\partial \mathcal{L}}{\partial \phi_i}^{k-1}_{avg}, \frac{\partial \mathcal{L}}{\partial \phi_i}^{k} \right) \right]. \quad (4)$$

The first and the second terms are the reconstruction error and the latent loss, respectively, and the last term is the gradient constraint.

We utilize anomaly detection as a validation framework to both qualitatively and quantitatively evaluate the performance of activation-based and gradient-based representations. In machine learning, the abnormal data is defined as data whose classes or attributes have not been learned during training. In the anomaly detection framework, images from one class of a dataset is considered as inliers and used for the training. Images from other classes are considered as outliers and both inliers and outliers are given to the network during the test time. The anomaly detection algorithms are expected to correctly classify both inliers and outliers by effectively distinguishing learned knowledge and not learned knowledge. Most of existing works for anomaly detection are based on the activation-based representations. In particular, they utilize reconstruction error or latent loss as measures for missing information and train probabilistic models on reconstructed image space or latent space to characterize anomalies (Zong et al., 2018; Pidhorskyi et al., 2018; Abati et al., 2018).

We perform two sets of experiments to thoroughly validate the effectiveness of the gradient-based representation. In the first experiment, we analyze the capability of activation-based and gradient-based representations in characterizing missing information. To be specific, we evaluate the anomaly detection performance of the proposed gradient loss defined by the gradient constraint along with the reconstruction error and the latent loss to show the potential in each representations for characterizing missing information. We use an autoencoder with 4-layer encoder and decoder throughout whole experiments. We train CAEs using mean squared error as the reconstruction error. Also, we train variational autoencoders (VAEs) using binary cross entropy as the reconstruction error and Kullback Leibler (KL) divergence as the latent loss (Kingma & Welling, 2013). For the gradient loss, we train both CAEs and VAEs with the gradient constraint defined as the last term in (4). Finally, we utilize these losses as abnormality scores and report the anomaly detection performance. In the second experiment, we develop an anomaly detection algorithm using **Grad**ient **Con**straint (GradCon) which detects anomalies by using the combination of reconstruction error and gradient loss as an abnormality score. We compare the performance of the proposed method with other benchmarking and state-of-the-art algorithms and show that GradCon outperforms all compared algorithms at least in one dataset.

We utilize four benchmarking datasets, which are CIFAR-10 (Krizhevsky & Hinton, 2009) , CURE-TSR (Temel et al., 2017), MNIST (LeCun et al., 1998), and fashion MNIST (fMNIST) (Xiao et al., 2017), to evaluate the performance of the proposed algorithm. CIFAR-10 dataset consists of 60,000 color images with 10 classes. CURE-TSR dataset has $637,560$ color traffic sign images which consist of 14 traffic sign types under 5 levels of 12 different challenging conditions. MNIST dataset contains 70,000 handwritten digit images from 0 to 9 and fMNIST dataset also has 10 classes of fashion products and there are 7,000 images per class. For CIFAR-10, CURE-TSR, and MNIST, we follow the protocol described in Perera et al. (2019) . We utilize the training and testing split of each dataset to conduct experiments and $10\%$ of training images are held out for validation. For fMNIST, we follow the protocol described in Pidhorskyi et al. (2018). The dataset is split into 5 folds and $60\%$ of each class is used for training, $20\%$ is used for validation, the remaining $20\%$ is used for testing. In experiments with CIFAR-10, MNIST, and fMNIST, we use images from

| | Plane | Car | Bird | Cat | Deer | Dog | Frog | Horse | Ship | Truck | Average |
|---|---|---|---|---|---|---|---|---|---|---|---|
| OCSVM | 0.630 | 0.440 | 0.649 | 0.487 | 0.735 | 0.500 | 0.725 | 0.533 | 0.649 | 0.508 | 0.586 |
| KDE | 0.658 | 0.520 | 0.657 | 0.497 | 0.727 | 0.496 | **0.758** | 0.564 | 0.680 | 0.540 | 0.610 |
| DAE | 0.411 | 0.478 | 0.616 | 0.562 | 0.728 | 0.513 | 0.688 | 0.497 | 0.487 | 0.378 | 0.536 |
| VAE | 0.634 | 0.442 | 0.640 | 0.497 | 0.743 | 0.515 | 0.745 | 0.527 | 0.674 | 0.416 | 0.583 |
| PixelCNN | **0.788** | 0.428 | 0.617 | 0.574 | 0.511 | 0.571 | 0.422 | 0.454 | 0.715 | 0.426 | 0.551 |
| GAN | 0.708 | 0.458 | 0.664 | 0.510 | 0.722 | 0.505 | 0.707 | 0.471 | 0.713 | 0.458 | 0.592 |
| AND | 0.735 | 0.580 | **0.690** | 0.542 | **0.761** | 0.546 | 0.751 | 0.535 | 0.717 | 0.548 | 0.641 |
| AnoGAN | 0.671 | 0.547 | 0.529 | 0.545 | 0.651 | 0.603 | 0.585 | 0.625 | 0.758 | 0.665 | 0.618 |
| DSVDD | 0.617 | **0.659** | 0.508 | 0.591 | 0.609 | **0.657** | 0.677 | **0.673** | 0.759 | **0.731** | 0.648 |
| OCGAN | 0.757 | 0.531 | 0.640 | **0.620** | 0.723 | 0.620 | 0.723 | 0.575 | **0.820** | 0.554 | 0.657 |
| **GradCon** | 0.760 | 0.598 | 0.648 | 0.586 | 0.733 | 0.603 | 0.684 | 0.567 | 0.784 | 0.678 | **0.664** |

Table 2: Anomaly detection results on CIFAR-10.

| | 0 | 1 | 2 | 3 | 4 | 5 | 6 | 7 | 8 | 9 | Average |
|---|---|---|---|---|---|---|---|---|---|---|---|
| OCSVM | 0.988 | **0.999** | 0.902 | 0.950 | 0.955 | 0.968 | 0.978 | 0.965 | 0.853 | 0.955 | 0.951 |
| KDE | 0.885 | 0.996 | 0.710 | 0.693 | 0.844 | 0.776 | 0.861 | 0.884 | 0.669 | 0.825 | 0.814 |
| DAE | 0.894 | **0.999** | 0.792 | 0.851 | 0.888 | 0.819 | 0.944 | 0.922 | 0.740 | 0.917 | 0.877 |
| VAE | 0.997 | **0.999** | 0.936 | 0.959 | 0.973 | 0.964 | 0.993 | 0.976 | 0.923 | 0.976 | 0.970 |
| PixelCNN | 0.531 | 0.995 | 0.476 | 0.517 | 0.739 | 0.542 | 0.592 | 0.789 | 0.340 | 0.662 | 0.618 |
| GAN | 0.926 | 0.995 | 0.805 | 0.818 | 0.823 | 0.803 | 0.890 | 0.898 | 0.817 | 0.887 | 0.866 |
| AND | 0.993 | **0.999** | **0.959** | 0.966 | 0.956 | 0.964 | **0.994** | 0.980 | **0.953** | **0.981** | **0.975** |
| AnoGAN | 0.966 | 0.992 | 0.850 | 0.887 | 0.894 | 0.883 | 0.947 | 0.935 | 0.849 | 0.924 | 0.913 |
| DSVDD | 0.980 | 0.997 | 0.917 | 0.919 | 0.949 | 0.885 | 0.983 | 0.946 | 0.939 | 0.965 | 0.948 |
| OCGAN | **0.998** | **0.999** | 0.942 | 0.963 | **0.975** | **0.980** | 0.991 | **0.981** | 0.939 | **0.981** | **0.975** |
| **GradCon** | 0.995 | **0.999** | 0.952 | **0.973** | 0.969 | 0.977 | **0.994** | 0.979 | 0.919 | 0.973 | 0.973 |

Table 3: Anomaly detection results on MNIST.

one class as inliers to train the network. During test, inlier images and the same number of oulier images randomly sampled from other classes are utilized. For CURE-TSR, challenge-free images are utilized as inliers to train the network. During test, challenge-free images are utilized as inliers and challenging version of these images are utilized as outliers. All the results are obtained using area under receiver operation characteristic curve (AUROC) and we also report F1 score in fMNIST dataset for the fair comparison with the state-of-the-art method (Pidhorskyi et al., 2018).

## 6 RESULTS

**Comparison between activation-based and gradient-based representations** We report the anomaly detection performance based on the gradient loss (Grad) along with reconstruction error (Recon) and the latent loss (Latent) using CIFAR-10 in Table 1. From this table, we can analyze three aspects of activation and gradient representations. First, by comparing the performance of CAE and CAE trained with gradient constraint (CAE + Grad), we analyze the effect of gradient constraint in characterizing missing information. The gradient constraint leads to a comparable average AUROC from reconstruction error and achieves the best performance from the gradient loss with an average AUROC of 0.661. Second, we evaluate the effect of latent constraint by comparing CAE and VAE. The latent loss achieves improved performance compared to the reconstruction error of CAE. However, the latent constraint imposed in the VAE sacrifices the average AUROC of reconstruction error by 0.042. Finally, comparison between VAE and VAE with gradient constraint (VAE + Grad) analyzes the effect of the gradient constraint with the activation constraint. The gradient loss in VAE + Grad achieves the second best performance by marginally sacrificing the average AUROC of reconstruction error and latent loss. From this experiment, we observe that imposing constraints on activation can degrade the performance of other activation-based representation in characterizing missing information. However, since gradients are obtained in parallel with activations, gradient constraint enables to achieve better performance in anomaly characterization by complementing the activation-based representations.

We perform anomaly detection in CURE-TSR to highlight the discriminant capability of gradient representation for diverse challenging conditions and levels. We compare the performance of CAE and CAE + Grad using the reconstruction error and the gradient loss. The goal of this task is to successfully detect whether given test image is affected by challenge condition or not. We report the performance for 8 challenging conditions and 5 levels in Fig. 4. Those challenge conditions are decolorization, lens blur, dirty lens, exposure, Gaussian blur, rain, snow, and haze as visualized in Fig. 4. For all challenging conditions and levels, CAE + Grad achieves the best performance. In particular, except for snow level 1∼3, the gradient loss achieves the best performance and for snow

level 1~3, the reconstruction error achieves the best performance. In terms of the average AUROC over challenge levels, the gradient loss of CAE + Grad outperforms the reconstruction error of CAE by the largest margin of $0.612$ in rain and the smallest margin of $0.089$ in snow. Overall, the gradient representation effectively characterizes challenge information that has not been learned and outperforms the activation-based representation.

**Comparison with the state-of-the-art methods** GradCon is the combination of the reconstruction error and the gradient loss from a CAE trained with the gradient constraint. We train CAEs by setting $\alpha = 0.03$ and $\Omega(z; \theta, \phi) = 0$ in (4). This loss utilized for training is utilized as an anomaly score during testing but with the increased $\alpha$ of $0.12$. We compare the proposed method with other algorithms including OCSVM (Schölkopf et al., 2001), KDE (Bishop, 2006), DAE (Hadsell et al., 2006), VAE (Kingma & Welling, 2013), PixelCNN (Van den Oord et al., 2016), GAN (Schlegl et al., 2017), AND (Abati et al., 2018), AnoGAN (Schlegl et al., 2017), DSVDD (Ruff et al., 2018), OC-GAN (Perera et al., 2019). The AUROC results on CIFAR-10 and MNIST are reported in Table 2 and Table 3, respectively. GradCon achieves the best average AUROC performance in CIFAR-10 while achieving comparable performance to the

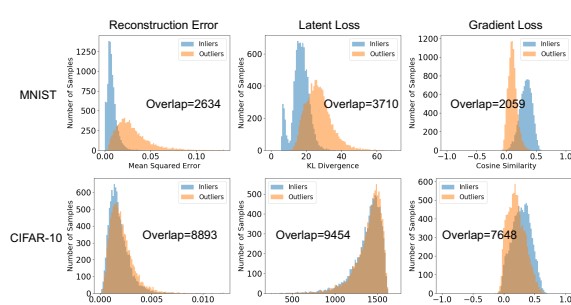

Figure 5: Histogram analysis on activation losses and gradient loss in MNIST and CIFAR-10.

| | % of outlier | 10 | 20 | 30 | 40 | 50 |
|---|---|---|---|---|---|---|
| F1 | GPND | **0.968** | **0.945** | 0.917 | 0.891 | 0.864 |
| | **GradCon** | 0.967 | **0.945** | **0.924** | **0.905** | **0.871** |
| AUC | GPND | 0.928 | 0.932 | 0.933 | 0.933 | 0.933 |
| | **GradCon** | **0.938** | **0.933** | **0.935** | **0.936** | **0.934** |

Table 4: Anomaly detection results on fMNIST.

best algorithms in MNIST. We note that while other state-of-the-art methods train additional deep networks on top of latent space or reconstructed image space to detect anomalies, GradCon is solely based on the gradient loss without training additional networks for anomaly detection. Therefore, the proposed method is computationally efficient compared to other methods. In Fig. 5, we visualize the distribution of reconstruction error, latent loss, and gradient loss for inliers and outliers to further elaborate the reasoning for state-of-the-art performance of the proposed method. We visualize each loss for all inliers and outliers from 10 classes in MNIST and CIFAR-10. In CIFAR-10, which contains color images with more complicated structure than MNIST, activation-based losses fail to effectively separate inliers and outliers. On the the other hand, the gradient loss still maintains the separation in CIFAR-10. The gradient loss achieves the least number of samples overlapped in both MNIST and CIFAR-10, which explains the state-of-the art performance achieved by GradCon in both datasets. We also evaluate the performance of GradCon in comparison with another state-of-the-art algorithm denoted as GPND (Pidhorskyi et al., 2018) in fMNIST. In this fMNIST experiment, we change the ratio of outliers in the test set from $10\%$ to $50\%$ and analyze the performance in terms of AUROC and F1 scores. The results in fMNIST are reported in Table 4. The proposed method outperforms GPND in all outlier ratios in terms of AUROC. Except for the $10\%$ of outlier ratio test set, the proposed method achieves higher F1 scores than GPND.

## 7 CONCLUSION

We propose a gradient-based representation for characterizing information that deep networks have not learned. We introduce our geometric interpretation of gradients and generalize it to high dimensional scenarios of deep learning through the proposed directional constraint on gradients. We also thoroughly evaluate the representation capability of gradients compared to that of activations. We validate the effectiveness of gradients in the context of anomaly detection and show that proposed method based on the gradient representation achieves the state-of-the-art performance in four benchmarking datasets. The experimental results show that the directional information of gradients effectively characterizes diverse missing information by complementing distance information from activations. Also, the gradient-based representation can provide a comprehensive perspective to handle data that cannot be represented by training data in diverse applications aiming to ensure the robustness of deep networks.

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
