# OpenReview forum: "Characterizing Missing Information in Deep Networks Using Backpropagated Gradients"
_ICLR.cc/2020/Conference — Reject_

### Official Review · AnonReviewer2 · 2019-10-27
**Official Blind Review #2**

**Rating:** 3

**Review:**

The authors present an interesting idea: creating representations based on gradients
with respect to the weights to supplement information missing from the training dataset.
The idea if very well motivated and gets the reader excited.
Their explanation is clear from a high-level, however, the paper as a whole
lacks rigorous justification. A lot of space is spent reviewing the role of gradients
in training, what reconstruction loss is, and the basics of back-propagation --- however,
many of their own propositions (e.g. why constrain gradients?) are given without reasoning.

The paper could be made more clear by giving details and by changing the flow. For instance,
Section 4 first gives a high-level explanation, then begins discussing specific experiments,
and then returns to the method and provides some details. Additionally, it isn't until page 6
(Section 5) that the authors introduce GradCon, one of the more promising ideas shared, this should
be a focal point in the paper and introduced early (and given more than two sentences for explanation).
The main proposal appears to be a modification of the loss function, but the paper may benefit from
discussing implementation details (for example, during training vs. testing).
Finally, the Figures (2 and 3 in particular) are not clear and need more explanation given in
the captions. Figure 3a and 3b tell an interesting story, but they're not easily digestable, nor is
the key take-away clear to the reader just by looking at the figure and caption.

The experimental results look reasonable and thorough, however the methods are sold on the
idea of better representations for data missing from the training set, whereas the results
are focused on anomaly detection. The method looks particularly promising at anomaly detection
tasks --- the authors may have a more clear paper if they focus on this aspect.
Overall, the paper presents a promising idea but it needs a more clear and rigorous presentation.


**Experience Assessment:**

I have read many papers in this area.

**Review Assessment: Checking Correctness Of Derivations And Theory:**

I carefully checked the derivations and theory.

**Review Assessment: Checking Correctness Of Experiments:**

I carefully checked the experiments.

**Review Assessment: Thoroughness In Paper Reading:**

I read the paper thoroughly.

---

### Official Review · AnonReviewer1 · 2019-10-28
**Official Blind Review #1**

**Rating:** 1

**Review:**

[Summary]
The paper proposed to use gradient-based presentation in deep neural networks to capture missing information unavailable in the activation-based network representation. It is claimed that the missing information that could not be encoded during learning from training set can be revealed by taking the gradient of an example with respect to the model parameters. Based on this idea, a new learning algorithm is proposed to combine both the conventional loss for activation-based presentation and gradient-based regularization. As an illustration, the method is evaluated on anomaly detection benchmarks with gradient-based representation as part of the anomaly inference.

[Comments]
I’m not sure if I fully understand the claimed contribution and how it is justified either theoretically and empirically. It seems to me that the paper claims the the current activation-based network does not fully encode information from training set during learning, and taking the gradient over a novel example (either training or test) with respect to (w.r.t,) model parameters can encode the missing information. I don't quite follow a few things, and had a difficult time in interpreting the novelty here.
- What is the missing information (a model should learn but fails to capture from training set) exactly? The example in the introduction uses digit 0 for training and digit 6 for testing and claims the vertical edge in 6 is missing. But isn’t that expected if the model is only trained with digit 0s only? How would one expect the model to learn the vertical edge if there is none in the training data?
- More critically, the idea of taking the gradient of a model w.r.t.its parameters over an example to encode geometric relationship of the example in the data manifold is by no means novel at the conceptual level. There have been at least a bunch of similar classical methods derived from the perspective of information geometry. For instance, the Fisher kernel and vector  have been well studied for more than a decade, with both theoretical treatment (e.g., “Exploiting Generative Models in Discriminative Classifiers” in NIPS 1998) and application across multiple areas (e.g., “Fisher Kernels on Visual Vocabularies for Image Categorization” in CVPR 2007). I don’t see any novelty here by doing this to a new model (deep neural networks), and it is frustrating to see classical methods are being “invented” over and over again by decorating them with fashionable wrappers without any reference to the origin in the literature.
- Besides, to use gradients to capture missing information, the paper proposes to append a regularization term by enforcing the gradient at particular learning iteration to be close to the mean of those of previous iterations (via cosine similarity) as in (3). This seems not quite different from many existing popular inertia-based strategies like gradients with momentum, Adam, etc. It needs to be elucidated more how the proposed optimizer compares to these benchmark methods.

In terms of implementation, I also found some details missing. For instance, the objective function of (4) requires evaluation of second order derivatives (since the second term in the cosSIM involves dL/dPhi(x, Phi)). How to efficiently compute this term is not clear to me yet.   It is also mentioned that only decoder of the AE is used for constraint gradient, The reason for this is also not explained.

For evaluation, only anomaly detection examples are provided. It would be great if more general tasks like image classification can be studied. After all, the proposed method is claimed to be a fairly general framework (the introduction actually uses image classification as the example), and evaluation on the most popular benchmarks provides the most convincing justification. Even on the anomaly detection tasks, the proposed approach does not seem to have solid advantages over baselines. I’m not sure how significant are the differences of 0.007 between GradCon and OCGAN (table 2 and 3), and those less than 1% in table 4.

Overall the paper does not seem to have a clear and well-defined motivation, contribution is also vague and trivial compared to literature, plus very convincing empirical justification. Thus I do not think it is ready for publication at ICLR.


**Experience Assessment:**

I have read many papers in this area.

**Review Assessment: Checking Correctness Of Derivations And Theory:**

I assessed the sensibility of the derivations and theory.

**Review Assessment: Checking Correctness Of Experiments:**

I assessed the sensibility of the experiments.

**Review Assessment: Thoroughness In Paper Reading:**

I read the paper at least twice and used my best judgement in assessing the paper.

---

### Official Review · AnonReviewer3 · 2019-10-29
**Official Blind Review #3**

**Rating:** 3

**Review:**

The authors consider gradients of some loss as a feature representation. The intro starts describing a form of classification loss, but this is switched to a GAE unsupervised loss in Eq. 1. In that section the authors somewhat show that a naive gradient representation does not work well, but at the same time show that it /may/ work with their method, introduced right after, and which consists in an "orthogonality" regularizer, introduced with any arbitrary loss later in Eq.3, and back to GAE in Eq.4. The authors evaluate whether this regularizer hurst reconstruction error

I have found this paper interesting, but confusingly presented and still in a development phase (i.e. good idea, but not properly exploited yet). The premise on which the paper builds is that gradient information is more relevant than activations. Yet, this is only shown to be true if that gradient information is "bent" to follow a given orthogonality constraint in a fairly heuristic way (Eq. 4). Many choices are going under the hood:

"We train CAEs by setting α = 0.03 and Ω(z; θ, φ) = 0 in (4). This loss utilized for training is utilized as an anomaly score during testing but with the increased α of 0.12"

The sentence above is quite mysterious. The framework is tested with an anomaly detection task, which is a bit underwhelming, and to be frank, the results in Tables 3 or 4 (MNIST / CIFAR10) are not particularly convincing, specially when one factors the fact that α has been strangely changed.

Because of this, I feel that overall the paper starts with a nice idea but has not reached yet the state in which it deserves being published.

other comments:
- I think it would help to formalize from the introduction what kind of gradient you are specifically referring to. Is it the gradient of the loss XE of the classifier w.r.t. an arbitrary label? although this is clarified later in the paper, the introduction is not informative enough (Fig. 1 is nice, but a proper statement is needed). In particular, gradients is presented as an alternative to Jacobians, but shouldn't it be some form of Jacobian (or collection of gradients)
- in Eq. 1, why use *both* z and f_theta(x) in the same line, since they are equal?
- Around Eq. 1, "The training is performed by minimizing the loss function, J(x; θ, φ), defined as follows". This is inacurate, J is not minimized to compute theta and phi: the sum over J(x_i; ...) is minimized. J is just an examplar loss. You minimize the expectation of J.
- "can be calculated through the backpropagation" --> remove "the"
- Your narrative in bottom of p.3 from "We visualize the" is a bit flawed: one could imagine an example where x_out ~= \hat{x}_out, i.e. for which the loss L is small but the regularizer is large, or inversely one in which J is dominated by a small regularizer loss but where the loss L is large. So associating the size of the gradient of J to the departure from the manifold is misleading. It seems however that this is corrected later in p.5, in which you now allude only to d \mathcal{L}
- aren't there analogies with these gradients and Fisher score vectors in parametric stats? i.e. d log p_theta(x) / d theta ?
- add more caption to Fig.3 describing what's going on, or at the very least a direct reference on the section / location in which the experiments are described.
- "with our proposed method which will be described in the following section" : I would avoid this kind of "movie" narrative which tries to create a "cinematic buildup". Stick to scientific order, i.e. describe first method, test and experiment with it next.
- "is the primal loss for a given task" --> supervised or unsupervised task?

**Experience Assessment:**

I have read many papers in this area.

**Review Assessment: Checking Correctness Of Derivations And Theory:**

I assessed the sensibility of the derivations and theory.

**Review Assessment: Checking Correctness Of Experiments:**

I assessed the sensibility of the experiments.

**Review Assessment: Thoroughness In Paper Reading:**

I read the paper at least twice and used my best judgement in assessing the paper.

---

### Decision · Program_Chairs · 2019-12-19

**Decision:**

Reject

**Comment:**

This paper proposes the use of gradient of the loss evaluated at the example with respect to the model parameters as the feature representation of that example. The authors performed an empirical analysis on anomaly detection benchmarks to demonstrate the practical benefits of the proposed method. While the reviewers find the idea interesting, the consensus is that the proposed method lacks justification, and that the main claims were not substantiated. While the reviewers proposed several key points of improvement, the raised issues were not addressed in the rebuttal. I will hence recommend rejection of this paper.